# LaKD: Length-agnostic Knowledge Distillation for Trajectory Prediction with Any Length Observations

**Yuhang Li**
Beijing Institute of Technology
596983629@qq.com

**Changsheng Li** *
Beijing Institute of Technology
lcs@bit.edu.cn

**Ruilin Lv**
Beijing Institute of Technology
3220231454@bit.edu.cn

**Rongqing Li**
Beijing Institute of Technology
lirongqing99@gmail.com

**Ye Yuan**
Beijing Institute of Technology
yuan-ye@bit.edu.cn

**Guoren Wang**
Beijing Institute of Technology
wanggrbit@126.com

## Abstract

Trajectory prediction is a crucial technology to help systems avoid traffic accidents, ensuring safe autonomous driving. Previous methods typically use a fixed-length and sufficiently long trajectory of an agent as observations to predict its future trajectory. However, in real-world scenarios, we often lack the time to gather enough trajectory points before making predictions, e.g., when a car suddenly appears due to an obstruction, the system must make immediate predictions to prevent a collision. This poses a new challenge for trajectory prediction systems, requiring them to be capable of making accurate predictions based on observed trajectories of arbitrary lengths, leading to the failure of existing methods. In this paper, we propose a **L**ength-**a**gnostic **K**nowledge **D**istillation framework, named **LaKD**, which can make accurate trajectory predictions, regardless of the length of observed data. Specifically, considering the fact that long trajectories, containing richer temporal information but potentially additional interference, may perform better or worse than short trajectories, we devise a dynamic length-agnostic knowledge distillation mechanism for exchanging information among trajectories of arbitrary lengths, dynamically determining the transfer direction based on prediction performance. In contrast to traditional knowledge distillation, LaKD employs a unique model that simultaneously serves as both the teacher and the student, potentially causing knowledge collision during the distillation process. Therefore, we design a dynamic soft-masking mechanism, where we first calculate the importance of neuron units and then apply soft-masking to them, so as to safeguard critical units from disruption during the knowledge distillation process. In essence, LaKD is a general and principled framework that can be naturally compatible with existing trajectory prediction models of different architectures. Extensive experiments on three benchmark datasets, Argoverse 1, nuScenes and Argoverse 2, demonstrate the effectiveness of our approach.

---

*Changsheng Li (lcs@bit.edu.cn) is the corresponding author

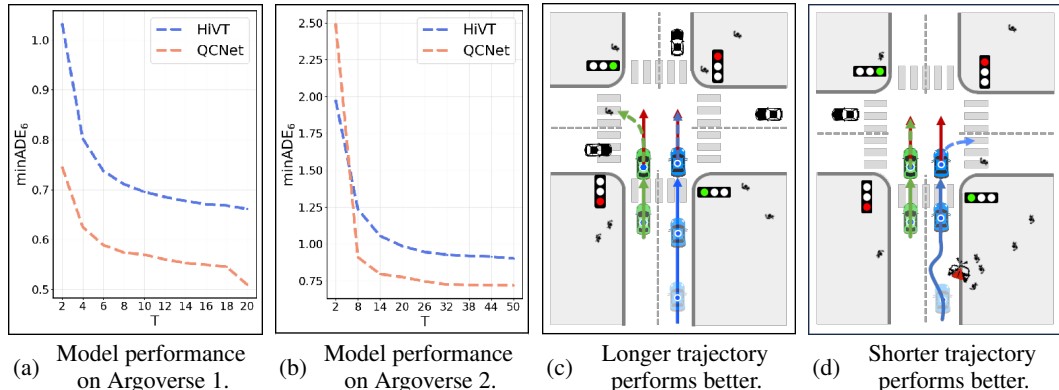

|     |                                          |     |                                          |     |                                      |     |                                      |
| --- | ---------------------------------------- | --- | ---------------------------------------- | --- | ------------------------------------ | --- | ------------------------------------ |
| (a) | Model performance on Argoverse 1.        | (b) | Model performance on Argoverse 2.        | (c) | Longer trajectory performs better.   | (d) | Shorter trajectory performs better.  |

Figure 1: Figure 1(a) and Figure 1(b) show the prediction results of HiVT [60] and QCNet [59] on the Argoverse 1 [4] and Argoverse 2 [51] datasets by using observed trajectories of different lengths, respectively. Figure 1(c) and Figure 1(d) display scenarios where longer trajectories perform better and shorter trajectories perform better, respectively. The red line represents the ground-truth future trajectory. The solid green and blue lines depict the observed trajectories, while the dashed green and blue lines illustrate the predicted trajectories.

## 1 Introduction

Predicting the future trajectories of dynamic agents in traffic scenarios is a critical task in autonomous driving, enabling autonomous vehicles to make safe decisions [55, 18]. Recently, numerous learning-based methods [60, 43, 39, 48, 50, 57, 59] have been proposed and have demonstrated their effectiveness in trajectory prediction tasks. These methods typically rely on fixed-length and sufficiently long historical trajectories as observations for accurately predicting future trajectories. However, In real-world scenarios, there is often insufficient time to gather an adequate number of observed trajectory points. For example, when a car suddenly appears around a corner, the trajectory prediction model needs to immediately make predictions by utilizing a small number of observed trajectory points to avoid collisions. This poses a new and challenging problem for trajectory prediction, requiring models to make accurate predictions based on observed trajectories of arbitrary lengths. However, as the number of observed trajectory points decreases, the performance of existing methods declines significantly, as shown in Figures 1(a) and 1(b). Therefore, it is essential to investigate models capable of handling observed trajectories of arbitrary lengths to accurately predict future trajectories.

In this paper, we propose a new knowledge distillation framework, **L**ength-**a**gnostic **K**nowledge **D**istillation, called **LaKD**, for trajectory prediction with observations of arbitrary lengths. Firstly, we note that longer trajectories often contain more temporal information, which can potentially lead to higher prediction accuracy compared to shorter trajectories. As shown in Figure 1(c), the blue vehicle's straight trajectory history can boost confidence in predicting continued straight paths. However, as the number of observed trajectory points increases, additional interference might be introduced. As depicted in Figure 1(d), despite the longer trajectory of the blue vehicle, it encompasses significant interference, leading to less accurate predictions compared to shorter trajectories. Inspired by this, we devise a dynamic length-agnostic knowledge distillation strategy to adaptively transfer knowledge among trajectories of different lengths. As we know, Knowledge Distillation (KD) techniques [2, 14] have been widely applied in various domains, including computer vision [7, 11], natural language processing [31, 12], etc. The basic idea of traditional KD algorithms is to optimize a smaller student model by distilling knowledge from a larger teacher model. In contrast to these KD methods, our strategy emphasizes dynamic knowledge transfer among trajectories of varying lengths, rather than the conventional KD of transferring knowledge from the teacher model to the student model. Our method shares a unique encoder for all trajectories of varying lengths to learn the latent representations of trajectories of varying lengths. It aims to distill the knowledge of 'good' trajectory features to 'bad' trajectories, with the assessment of 'good' or 'bad' trajectories based on their prediction performance. This strategy facilitates adaptive knowledge exchange between long and short trajectories. It aids long trajectories in filtering out interfering information and assists

short trajectories in capturing richer temporal information, ultimately obtaining the optimal feature representation for predicting the agent's trajectory.

It is worth noting that utilizing a single encoder as both the teacher and student models may affect the prediction performance of a 'good' trajectory when distilling from a 'good' trajectory to a 'bad' trajectory, leading to knowledge collision during the distillation process. A straightforward solution is to train a separate encoder for each trajectory length, but this approach significantly increases computational complexity. To address this issue, we devise a dynamic soft-masking strategy. Since different neuron units in a neural network model usually play different roles for different input data [37], the core idea of our strategy is to perform soft-masking on the neuron units during gradient updates. Specifically, when training on a 'good' observation trajectory, the importance of the neuron units in the network is calculated based on the gradients. Subsequently, during length-agnostic knowledge distillation, gradients of crucial neuron units are multiplied by a lower update weight to mitigate significant updates. Conversely, less important units' gradients are multiplied by a higher update weight to prioritize their updates. Through this approach, knowledge conflicts can be effectively resolved during the knowledge distillation process.

Our contributions can be summarized as follows: (1) We propose LaKD, a length-agnostic knowledge distillation framework for trajectory prediction with observations of any length. LaKD is plug-and-play and compatible with existing models, enabling them to gracefully handle observed trajectories of arbitrary lengths. (2) We design a new knowledge distillation strategy that dynamically transfers knowledge among trajectories of varying lengths. This approach helps long trajectories filter out interfering information and enables short trajectories to capture richer temporal details. Additionally, we devise a dynamic soft-masking strategy to protect crucial neuron units from disruption and prevent knowledge collision during transfer. (3) We perform extensive experiments on three widely-used benchmark datasets, and demonstrate that LaKD significantly outperforms the baselines. Moreover, we show the compatibility of LaKD by integrating it with different trajectory prediction models.

## 2    Related Works

**Traditional Trajectory Prediction.**    Traditional trajectory prediction methods aim to predict future trajectories of agents given sufficiently long observed trajectories. To date, many methods have been proposed, including coordinate system based methods [50, 17], interactive behavior modeling based methods [28, 27, 25], multimodal approaches [44, 46]. The representative works among coordinate system based methods are pairwise-relative [60, 8, 17, 59, 57], which can simultaneous predict trajectories for multiple agents while reducing memory consumption and inference latency. Meanwhile, interaction behaviors play an important role in trajectory prediction. To model interactive behaviors within scenes, methods such as Graph Neural Networks [28, 27, 22, 5, 33, 41, 42] and attention mechanisms [39, 1, 6, 36, 32, 52, 23] are introduced. Given the substantial uncertainty surrounding road agents, researchers are exploring diverse methods by integrating multimodal information into predicted trajectories, such as GAN-based [29, 44, 46, 58, 16, 26, 9, 13], VAE-based [47, 49, 21], flow-based [30, 56], and diffusion models [10, 15, 24, 35] to generate multimodal trajectories. However, these methods generally perform well with fixed-length and sufficiently long historical trajectories but experience a significant performance drop when the length of observable historical trajectories varies.

**Instantaneous Trajectory Prediction.**    Recently, significant advances have been made in instantaneous pedestrian trajectory prediction tasks, using very short (i.e., two frames) historical trajectories. For example, MOE [45] introduces a unified feature extractor and a pre-training mechanism to capture effective information from momentary observations. DTO [38] employs a knowledge distillation technique to transfer knowledge from long trajectories to short ones. BCDiff [24] develops a bidirectional diffusion model that simultaneously generates both unobserved historical and future trajectories. However, when confronted with input data containing varying numbers of frames, they necessitate training a model for each case, resulting in limited generalizability and high computational complexity. In contrast to these works, we focus on studying how to perform length-agnostic knowledge distillation to adaptively transfer knowledge among long and short trajectories, so as to accurately predict future trajectories with observations of arbitrary lengths.

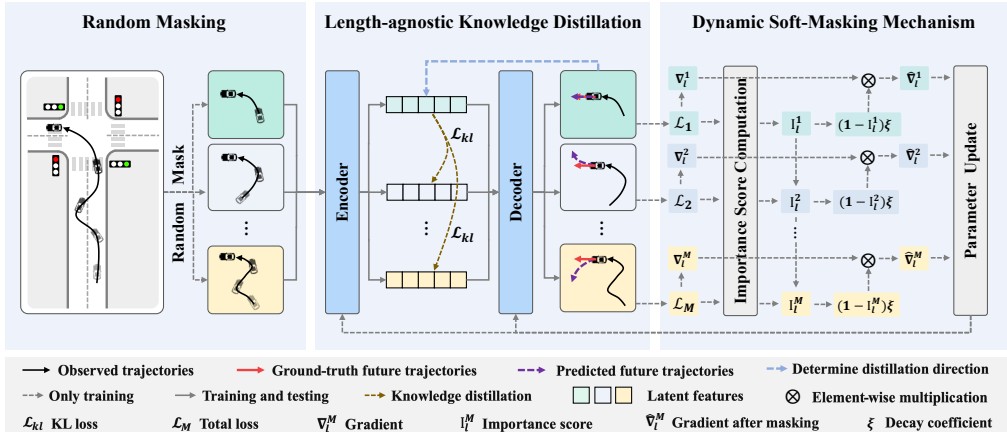

Figure 2: Illustration of our LaKD framework. During training, we randomly mask historical trajectories $M$ times to generate observed trajectories of varying lengths. Subsequently, we design a length-agnostic knowledge distillation module to dynamically transfer knowledge across trajectories of different lengths. Finally, we devise a dynamic soft-masking mechanism during gradient updates to effectively prevent knowledge conflicts. During inference, random masking, knowledge distillation, and dynamic soft-masking are not implemented.

**Trajectory Prediction with Complex Observations.** Currently, there are limited works focusing on complex observed trajectories for trajectory prediction. The recently proposed GC-VRNN framework [53] facilitates the concurrent execution of incomplete trajectory completion and prediction tasks in a unified framework. However, this model does not take into account important traffic information, e.g., lane, making it unsuitable for vehicle trajectory prediction tasks. The FLN framework [54], which is most closely related to our work, propagates long historical trajectory information into medium and short trajectories to optimize the fitting of invariant features across multiple subnetworks. However, this strategy requires maintaining three models simultaneously during training, sharply increasing computational complexity. In addition, it is plug-and-play but can only be integrated with Transformer based models. Moreover, this method assumes that longer observed trajectories always contain more useful information for trajectory prediction, and transfers knowledge from longer trajectories to shorter ones. Different from FLN, we observe that longer observed trajectories do not necessarily contain more valuable information than shorter ones for trajectory prediction, and thus explore a length-agnostic knowledge distillation to dynamically transfer knowledge among long and short trajectories, enabling our method to gracefully handle observed trajectories of arbitrary lengths.

## 3 Method

### 3.1 Problem Formulation

We denote the observed state sequence of the target agent as $\mathbf{X}^{obs} = \{x_1, x_2, ..., x_T\}$, where $T$ represents the observed time steps of the target agent, and it can be of arbitrary length greater than $1^2$. $x_i \in \mathbb{R}^2$ is the location of the agent at time step $i$. Additionally, we define the ground-truth future trajectories as $\mathbf{X}^{gt} = \{x_{T+1}, x_{T+2}, ..., x_{T+F}\}$, and the predicted future possible $K$ trajectories as $\widehat{\mathbf{X}} = \{(\hat{x}^k_{T+1}, \hat{x}^k_{T+2}, ..., \hat{x}^k_{T+F})\}_{k \in [1,K]}$, where $F$ denotes the length of the future trajectory. Our objective is to develop a flexible trajectory prediction method capable of handling the case of observed trajectories of arbitrary lengths. Given that longer trajectories contain richer temporal information yet may also entail additional interference, their performance relative to short trajectories can vary. Thus, we attempt to explore a length-agnostic knowledge distillation framework for dynamically transferring knowledge among long and short trajectories, enabling long trajectories to filter out interference and allowing short trajectories to capture richer temporal details. By doing so, we aim to enhance the performance of trajectory prediction with observations of any lengths.

---

$^2$Previous works [45, 24] have shown that when the agent has only one frame of historical trajectory data, it cannot be predicted due to the lack of basic information such as velocity and direction.

## 3.2 Overall Framework

The overall framework of the proposed LaKD is shown in Figure 2. Our framework consists of two parts: a length-agnostic knowledge distillation mechanism and a dynamic soft-masking strategy. First, to enhance the model's ability to handle observed trajectories $\mathbf{X}^{obs}$ of arbitrary lengths, we propose a length-agnostic knowledge distillation mechanism. This mechanism first evaluates the performance of the predicted trajectories $\widehat{\mathbf{X}}$, and then determines the direction of knowledge distillation accordingly. Finally, it promotes adaptive knowledge exchange among trajectories of varying lengths, helping long trajectories filter out interfering information and short trajectories capture richer temporal information. However, since this strategy uses a single encoder as both the teacher and student models, it risks causing knowledge conflicts during distillation. Therefore, we propose a dynamic soft-masking strategy to address this issue. Specifically, When training on a 'good' observation trajectory, the importance of neuron units in the network can be determined by their gradients. During length-agnostic knowledge distillation, crucial neuron gradients are multiplied by a lower update weight to mitigate significant updates, while less important gradients are multiplied by a higher update weight to prioritize their updates. This strategy can effectively resolve knowledge conflicts during distillation, such that our LaKD can effectively perform trajectory prediction based on observations of arbitrary lengths. In essence, LaKD is a plug-and-play approach that can be easily integrated with existing trajectory prediction models, enabling accurate predictions based on observed trajectories of varying lengths.

## 3.3 Length-agnostic Knowledge Distillation

In this section, we introduce our proposed length-agnostic knowledge distillation mechanism, which can facilitate information exchange among trajectories of different lengths, thereby enhancing the model's ability to handle observed trajectories of arbitrary lengths.

First, we obtain $\mathbf{X}^{obs}$ of $M$ different lengths by performing $M$ random masks on the same observed trajectory, where the $m$-th trajectory is denoted as $\mathbf{X}_m^{obs}$. As shown in Figure 2, these trajectories are fed into the backbone $\Phi$ to generate the latent features $\mathbf{V}_m$ and predicted trajectories $\widehat{\mathbf{X}}_m$:

$$\mathbf{V}_m = \Phi_E(\mathbf{X}_m^{obs}; \phi_E), \qquad \widehat{\mathbf{X}}_m = \Phi_D(\mathbf{V}_m; \phi_D), \tag{1}$$

where $\Phi_E$ and $\Phi_D$ denote the encoder and decoder of $\Phi$, with parameterized by $\phi_E$ and $\phi_D$, respectively. The backbone $\Phi$ can be any trajectory prediction model, e.g., HiVT [60] and QCNet [59] used in this paper, making our method plug-and-play.

As aforementioned, longer trajectories contain richer temporal information but may also involve additional interference for trajectory prediction, thus we design a length-agnostic knowledge distillation strategy, where knowledge transfer can occur from longer trajectories to shorter ones, as well as from shorter to longer trajectories. To dynamically determine the direction of knowledge transfer, we employ the prediction performance based on different observed trajectories to find a 'good' trajectory, and attempts to distill the knowledge embedded in its latent features $\mathbf{V}_m$ to those of 'bad' trajectories. To measure the prediction performance, we calculate the minimum distance $D_m$ between the predicted trajectories $\widehat{\mathbf{X}}_m$ and the ground-truth trajectories $\mathbf{X}^{gt}$ using the $l_2$ norm:

$$D_m = \min_{i \in \{1, 2, \ldots, k\}} \left( \sqrt{\sum_{j=T+1}^{T+F} \|\hat{x}_{ij} - x_j\|^2} \right). \tag{2}$$

During training, if the prediction performance of the current observation trajectory is worse than that of a previous 'good' observation trajectory, we begin to distill knowledge from the 'good' trajectory to the current trajectory. In this paper, we use the latent features as knowledge for transfer, and use the KL divergence [20] to minimize the following distillation loss to achieve the goal:

$$\mathcal{L}_{kl} = \mathrm{KL}(\mathbf{V}_m | \mathbf{V}_{good}), \tag{3}$$

where $\mathbf{V}_{good}$ represents the latent features of the 'good' trajectory. By Eq. (3), the features $\mathbf{V}_m$ of 'bad' trajectories are expected to be optimized towards those of 'good' trajectories, i.e., $\mathbf{V}_{good}$. This facilitates effective knowledge transfer from 'good' to 'bad' trajectories. It is worth noting that different from traditional knowledge distillation optimizing a smaller student model by distilling knowledge from a larger teacher model, we utilize a unique encoder to encode all trajectories of

arbitrary lengths, and distill knowledge from 'good' trajectory to 'bad' one. This may lead to knowledge collision during the distillation process, degrading the feature representation capability of the 'good' trajectory. To this end, we devise a dynamic soft-masking strategy to address the issue of knowledge collision.

### 3.4 Dynamic Soft-Masking

As we know, different neuron units in a neural network model typically play distinct roles for various input data [37]. Thus, we attempt to perform soft-masking on the neuron units during gradient updates. Specifically, when training on a 'good' observation trajectory, we determine the importance of the neuron units based on their gradients. During length-agnostic knowledge distillation, the gradients of crucial neuron units are multiplied by a lower update weight to prevent significant updates, while the gradients of less important units are multiplied by a higher update weight to prioritize their updates. By this strategy, knowledge conflicts can be effectively resolved during the distillation process.

**Importance Score of Neuron Unit.** During the training process, if the gradient of a neuron unit is large, it indicates that changing it will have a significant impact on the result [37, 19]. Building on this, we aim to identify which units are essential for the model to produce accurate predictions. To do so, we first calculate the importance scores of the different units in the network as follows:

$$\boldsymbol{I}_u = \frac{1}{B} \sum_{b=1}^{B} |\frac{\partial \mathcal{L}(\widehat{\mathbf{X}}_b, \mathbf{X}_b^{gt})}{\partial \boldsymbol{g}_u}|, \tag{4}$$

where $B$ denotes the batch size. $\widehat{\mathbf{X}}_b$ and $\mathbf{X}_b^{gt}$ represent the predicted trajectories and the ground-truth trajectories, respectively. $\boldsymbol{g}_u$ is introduced as a virtual parameter for calculating the importance $\boldsymbol{I}_u$ of units, where we fix $\boldsymbol{g}_u$ to 1 in the training. $\boldsymbol{I}_u$ is the importance score of the $u$-th neuron unit.

Since the gradients of the neuron units are usually very small, they cannot be directly applied to the calculation of soft masking weights. Therefore, it is necessary to confine the values of importance scores within the range [0,1]. To achieve this, we first normalize the importance scores of all units within each layer, ensuring a mean of 0 and a standard deviation of 1. Then, we apply the Tanh activation function to these normalized scores as:

$$\hat{\boldsymbol{I}}_u = (\tanh\left(\frac{\boldsymbol{I}_u - \mu}{\sigma}\right) + 1)/2, \tag{5}$$

where $\mu$ represents the average importance of all units in the $l$-th layer, while $\sigma$ denotes their variance.

**Accumulation of Importance Scores.** Due to the fact that different units in the model play varying roles for trajectories of varying lengths, it is necessary to preserve the model's ability as much as possible during training. Therefore, during the $m$-th training iteration, we need to comprehensively consider the importance of units from the previous $m$-1 training iterations, and employ the element-wise maximum (EMax) operation for calculating the cumulative importance $\hat{\boldsymbol{I}}_u^{(\leq m-1)}$ of the model up to the $(m$-1)-th iteration:

$$\hat{\boldsymbol{I}}_u^{(\leq m-1)} = \text{EMax}(\{\hat{\boldsymbol{I}}_u^{(m-1)}, \hat{\boldsymbol{I}}_u^{(\leq m-2)}\}), \tag{6}$$

where we set $\hat{\boldsymbol{I}}_u^{(0)}$ uniformly to 0.

**Dynamic Soft-Masking of Units.** During early training stages, when the model's predictive capability is initially constrained, the informativeness of unit importance scores is limited. As training progresses, the reliability of these scores gradually improves. Therefore, we introduce a dynamic decay coefficient $\xi$ to control the strength of the soft-masking. The specific formulas for the decay coefficient and the dynamic soft-masking mechanism based on the importance of units are as follows:

$$\hat{\nabla}_u = (1 - \hat{\boldsymbol{I}}_u^{(\leq m-1)} * \xi) \otimes \nabla_u, \tag{7}$$

$$\xi = \begin{cases} \min(\mathcal{L}_{reg} * \gamma, 1) & \text{if } \mathcal{L}_{reg} < 0 \\ 0 & \text{otherwise,} \end{cases} \tag{8}$$

where $\gamma$ is a hyperparameter. $\mathcal{L}_{reg}$ represents the regression loss between the predicted trajectories and the ground-truth trajectories, as used in HiVT [60] and QCNet [59]. $\nabla_u$ and $\hat{\nabla}_u$ represent the gradients of the units before and after soft-masking, respectively. The dynamic soft-masking mechanism effectively addresses the issue of knowledge conflict between trajectories of different lengths, promotes cross-length information exchange and thereby enhances the model's ability to predict trajectories based on observations of arbitrary lengths.

### 3.5 Optimization and Inference

**Optimization.** Following HiVT [60] and QCNet [59], we also adopt the negative log-likelihood as the regression loss $\mathcal{L}_{reg}$, which regresses the trajectory closest to the ground truth. In addition, We also use the cross-entropy loss as the classification loss $\mathcal{L}_{cls}$ to optimize the trajectory prediction model. Finally, the total loss function can be expressed as follows:

$$\mathcal{L} = \mathcal{L}_{reg} + \alpha \mathcal{L}_{cls} + \beta \mathcal{L}_{kl}, \tag{9}$$

where $\alpha$ and $\beta$ are the hyperparameters used to balance the contributions of different loss functions. We provide the pseudo-code of the training procedure in Appendix A.1.

**Inference.** After training, the model can be utilized for trajectory prediction based on observations of arbitrary lengths. For a new observed trajectory of any length, we directly input it into the encoder and decoder for future trajectory prediction, bypassing knowledge distillation and soft masking.

## 4 Experiments

### 4.1 Experimental Settings

**Dataset.** We evaluate the performance of our method on three widely used datasets: Argoverse 1 [4], nuScenes [3] and Argoverse 2 [51]. The Argoverse 1 dataset comprises 323,557 real driving scenes from Miami and Pittsburgh. The observation duration is 5 seconds with a sampling frequency of 10Hz. Traditional trajectory prediction approaches typically assume that the first 2 seconds represent the historical observed trajectories, while the last 3 seconds are considered as the future ground-truth trajectories. The nuScenes dataset comprises 32,186 training scenarios, 8,560 validation scenarios, and 9,041 test scenarios. Each scenario spans 8 seconds, sampled at 2 Hz. Traditional trajectory prediction approaches typically assume that the first 2 seconds (5 locations) are used as the observed trajectory, while the last 6 seconds are designated as the future ground-truth trajectory. The Argoverse 2 dataset includes 250,000 scenes spanning across six cities. The observation duration is 11 seconds with a sampling frequency of 10Hz. Traditional trajectory prediction approaches typically assume that the first 5 seconds are used as historical observed trajectories, while the last 6 seconds serve as future ground-truth trajectories. By masking trajectories on these datasets, we aim to evaluate the effectiveness of our trajectory prediction method with observations of arbitrary lengths.

**Evaluation Metrics.** To comprehensively evaluate the model, we employ a set of evaluation metrics based on the minimum Average Displacement Error ($\mathrm{minADE}$), minimum Final Displacement Error ($\mathrm{minFDE}$), and Miss Rate ($\mathrm{MR}$) as:

$$\mathrm{min}\overline{\mathrm{ADE}}_K = \frac{1}{H-1} \sum_{i=2}^{H} (\mathrm{minADE}_K^{T=i}), \tag{10}$$

$$\mathrm{min}\overline{\mathrm{FDE}}_K = \frac{1}{H-1} \sum_{i=2}^{H} (\mathrm{minFDE}_K^{T=i}), \tag{11}$$

$$\overline{\mathrm{MR}}_K = \frac{1}{H-1} \sum_{i=2}^{H} (\mathrm{MR}_K^{T=i}), \tag{12}$$

where $H$ denotes the maximum number of observation points, and $K$ represents the number of trajectories to be predicted. $T = i$ represents the number of observation points. We evaluate the performance for each observation length and then average the results across all lengths to obtain the final outcome.

Table 1: Comparisons of different methods on Argoverse 1 and Argoverse 2, evaluated using $\overline{\text{minADE}}$, $\overline{\text{minFDE}}$ and $\overline{\text{MR}}$ metrics. The best results are highlighted in bold.

| Dataset | Methods | K=1 | | | K=6 | | |
|---|---|---|---|---|---|---|---|
| | | $\overline{\text{minADE}}$ | $\overline{\text{minFDE}}$ | $\overline{\text{MR}}$ | $\overline{\text{minADE}}$ | $\overline{\text{minFDE}}$ | $\overline{\text{MR}}$ |
| Argoverse 1 | HiVT-Orig | 1.4733 | 3.1834 | 0.5267 | 0.7255 | 1.0740 | 0.1124 |
| | HiVT-RM | 1.4189 | 3.0599 | 0.5104 | 0.7070 | 1.0447 | 0.1053 |
| | HiVT-DTO | 1.3999 | 3.0262 | 0.5056 | 0.7032 | 1.0350 | 0.1039 |
| | HiVT-FLN | 1.4011 | 3.0288 | 0.5051 | 0.7026 | 1.0325 | 0.1033 |
| | HiVT-LaKD | **1.3317** | **2.8799** | **0.4901** | **0.6807** | **0.9864** | **0.0928** |
| Argoverse 1 | QCNet-Orig | 1.1656 | 2.4021 | 0.3860 | 0.5791 | 0.7399 | 0.0734 |
| | QCNet-RM | 1.0995 | 2.2550 | 0.3630 | 0.5684 | 0.7115 | 0.0703 |
| | QCNet-DTO | 1.0708 | 2.2303 | 0.3563 | 0.5418 | 0.6848 | 0.0671 |
| | QCNet-FLN | 1.0631 | 2.2083 | 0.3579 | 0.5411 | 0.6680 | 0.0671 |
| | QCNet-LaKD | **0.9982** | **2.0718** | **0.3439** | **0.5240** | **0.6581** | **0.0640** |
| nuScenes | HiVT-Orig | 3.5973 | 8.3062 | 0.8518 | 1.5289 | 2.8261 | 0.4377 |
| | HiVT-RM | 3.6580 | 8.4889 | 0.8647 | 1.5245 | 2.8068 | 0.4716 |
| | HiVT-DTO | 3.5860 | 8.2556 | 0.8514 | 1.5105 | 2.7379 | 0.4350 |
| | HiVT-FLN | 3.5640 | 8.1928 | 0.8488 | 1.5094 | 2.7489 | 0.4427 |
| | HiVT-LaKD | **3.4296** | **7.8882** | **0.8369** | **1.4793** | **2.6934** | **0.4329** |
| nuScenes | QCNet-Orig | 4.3134 | 9.7857 | 0.8588 | 1.4719 | 2.5831 | 0.4600 |
| | QCNet-RM | 4.1723 | 9.4672 | 0.8622 | 1.5255 | 2.6303 | 0.4611 |
| | QCNet-DTO | 4.1447 | 9.4552 | 0.8580 | 1.4653 | 2.5798 | 0.4317 |
| | QCNet-FLN | 4.1169 | 9.3639 | 0.8562 | 1.4676 | 2.5448 | 0.4344 |
| | QCNet-LaKD | **4.0663** | **9.2524** | **0.8523** | **1.4594** | **2.4901** | **0.4023** |
| Argoverse 2 | HiVT-Orig | 2.5502 | 6.5586 | 0.7455 | 1.0561 | 2.1093 | 0.3275 |
| | HiVT-RM | 2.2848 | 6.0548 | 0.7249 | 0.9457 | 1.9283 | 0.2994 |
| | HiVT-DTO | 2.2769 | 6.0548 | 0.7275 | 0.9324 | 1.8946 | 0.2903 |
| | HiVT-FLN | 2.2786 | 6.0464 | 0.7240 | 0.9287 | 1.8838 | 0.2891 |
| | HiVT-LaKD | **2.2066** | **5.8769** | **0.7161** | **0.9183** | **1.8686** | **0.2791** |
| Argoverse 2 | QCNet-Orig | 2.1006 | 5.2219 | 0.6299 | 0.8339 | 1.3849 | 0.1884 |
| | QCNet-RM | 1.7452 | 4.4404 | 0.5957 | 0.7508 | 1.3184 | 0.1671 |
| | QCNet-DTO | 1.7713 | 4.4900 | 0.5979 | 0.7454 | 1.2924 | 0.1671 |
| | QCNet-FLN | 1.6940 | 4.2373 | 0.5808 | 0.7370 | 1.2595 | 0.1596 |
| | QCNet-LaKD | **1.6574** | **4.1505** | **0.5753** | **0.7258** | **1.2420** | **0.1555** |

**Backbone and Baselines.** To demonstrate the compatible ability of our LaKD, we combine it with two representative trajectory prediction methods: **HiVT** [60] and **QCNet** [59]. To verify the effectiveness of our method, we compare LaKD with FlexiLength Network (**FLN**) [54], the work most related to ours. FLN integrates trajectory data with diverse observation lengths and attempts to learn temporally invariant representations for future trajectory predictions. We also compare LaKD with (**DTO**) [38]. DTO is initially developed for instantaneous trajectory prediction. To ensure fairness, we modify its framework to distill from complete trajectories into arbitrary length trajectories. Moreover, we take **Orig** and **RM** as our baselines. **Orig** denotes using the original fixed-length observed trajectories as inputs for training the backbones, while **RM** involves randomly masking the original observed trajectories to generate trajectories of varying lengths as inputs for training the backbones.

**Implementation Details.** During training, we set $M$ in our LaKD to 3, and both $\alpha$ and $\beta$ to 1. For Argoverse 1, nuScenes, and Argoverse 2, we set $\gamma$ to -1, -0.65, and -1.35, respectively. The dimensionality of the encoded latent feature $\mathbf{V}^m$ is set to 128. We utilize the AdamW optimizer [34], setting the learning rate and weight decay parameters to 5e-4 and 1e-4, respectively. The batch size is set to 32. The experiments are implemented using PyTorch [40] on the NVIDIA GeForce RTX 4090.

## 4.2 Results and Analysis

**Performance on Trajectory Prediction with Observations of Arbitrary Lengths.** We evaluate the overall performance of our method, as listed in Table 1. Based on Table 1, our method outper-

forms all others, particularly surpassing FLN, across all the three datasets. This demonstrates the effectiveness of our method in predicting future trajectories from observations of varying lengths. Moreover, our LaKD outperforms **Orig**, indicating the necessity of developing a trajectory prediction method specifically designed to handle observations of varying lengths. Finally, our LaKD achieves the best performance with various backbones, demonstrating the compatibility of our method. More detailed results are presented in Appendix A.2.

**Ablation Study.** We conduct ablation studies to validate the effectiveness of our proposed components using HiVT as the backbone on the Argoverse 1 dataset. Since the Random Masking strategy was first presented in this paper, we also conduct ablation study to verify its effectiveness. The results are shown in Table 2. The experiments demonstrate that as we progressively remove the dynamic soft-masking mechanism (DSM), the Length-agnostic Knowledge Distillation (LaKD), and the Random Masking (PM), the performance of our method gradually declines, demonstrating the effectiveness of our proposed components. By combining these components, our method achieves the best performance.

Table 2: Ablation study of our method on the Argoverse 1 dataset.

| RM | LaKD | DSM | K=1 | | | K=6 | | |
|----|------|-----|-----------------|-----------------|-------------------|-----------------|-----------------|-------------------|
| | | | $\mathrm{min\overline{ADE}}$ | $\mathrm{min\overline{FDE}}$ | $\overline{\mathrm{MR}}$ | $\mathrm{min\overline{ADE}}$ | $\mathrm{min\overline{FDE}}$ | $\overline{\mathrm{MR}}$ |
| | | | 1.4733 | 3.1834 | 0.5267 | 0.7255 | 1.0740 | 0.1124 |
| ✓ | | | 1.4189 | 3.0599 | 0.5104 | 0.7070 | 1.0447 | 0.1053 |
| ✓ | ✓ | | 1.3619 | 2.9511 | 0.5051 | 0.6851 | 0.9965 | 0.0948 |
| ✓ | ✓ | ✓ | **1.3317** | **2.8799** | **0.4901** | **0.6807** | **0.9864** | **0.0927** |

**Analysis of Different Mask Numbers $M$.** We investigate the impact of different mask numbers $M$ in our LaKD on the trajectory prediction performance. We use HiVT [60] as the backbone, and list the results in Table 3. Since our method involves randomly masking historical trajectories $M$ times in each training iteration and continues for a sufficient number of epochs, observation trajectories of all different lengths are seen during training, regardless of the value of $M$. Consequently, the model's performance does not significantly degrade as $M$ changes, indicating that the model is not sensitive to $M$. This makes $M$ easy to set in real-world scenarios. For our experiments, we set $M = 3$.

Table 3: Analysis of our method with different $M$ on the Argoverse 1 dataset.

| M | K=1 | | | K=6 | | |
|---|-----------------|-----------------|-------------------|-----------------|-----------------|-------------------|
| | $\mathrm{min\overline{ADE}}$ | $\mathrm{min\overline{FDE}}$ | $\overline{\mathrm{MR}}$ | $\mathrm{min\overline{ADE}}$ | $\mathrm{min\overline{FDE}}$ | $\overline{\mathrm{MR}}$ |
| 2 | 1.3457 | 2.9116 | 0.4943 | 0.6808 | 0.9863 | 0.0930 |
| 3 | **1.3317** | **2.8799** | **0.4901** | **0.6807** | **0.9864** | **0.0928** |
| 4 | 1.3414 | 2.9017 | 0.4938 | 0.6814 | 0.9867 | 0.0934 |
| 5 | 1.3563 | 2.9359 | 0.5010 | 0.6851 | 0.9973 | 0.0961 |
| 6 | 1.3486 | 2.9198 | 0.4977 | 0.6878 | 1.0025 | 0.0943 |

**Qualitative Analysis.** To intuitively demonstrate the effectiveness of our LaKD, we perform a qualitative experiment on the Argoverse 2 dataset, as shown in Figure 3. The first row features a scenario at a T-junction where the agent is about to turn, with an observed trajectory spanning 5 points. The second row illustrates a scenario at a fork in the road, where the agent is preparing to change lanes, with an observed trajectory of 10 points. It is observable that across different scenarios, our method exhibits higher accuracy compared to other models.

## 5   Conclusion

In this paper, we propose a length-agnostic knowledge distillation framework for trajectory prediction with observations of any length. This framework enables long trajectories to filter out interfering information and short trajectories to capture richer temporal details. To address knowledge conflicts during distillation, we devise a dynamic soft-masking mechanism to protect crucial neuron units from disruption, thereby enhancing prediction performance. Extensive experiments on the Argoverse 1, nuScenes, and Argoverse 2 datasets demonstrate the effectiveness of our approach and its compatibility with various trajectory prediction models.

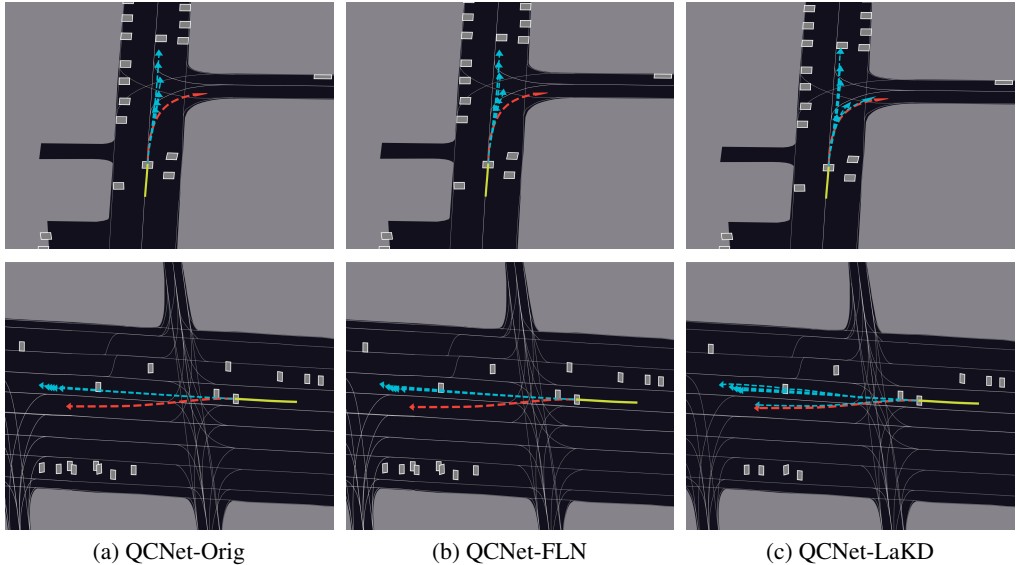

|     (a) QCNet-Orig     |     (b) QCNet-FLN     |     (c) QCNet-LaKD     |

Figure 3: Qualitative results on the Argoverse 2 dataset using (a) QCNet-Orig, (b) QCNet-FLN, and (c) QCNet-LaKD. The observed trajectories, ground-truth trajectories and predicted trajectories are shown in green, red and blue, respectively. Our predicted future trajectories are closer to the ground-truth, compared to other methods.

## Acknowledgments and Disclosure of Funding

This work was supported by the NSFC under Grants 62122013, U2001211.

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

# A Appendix

## A.1 Training Procedure of LaKD

We present the training procedure for LaKD in Algorithm 1.

---
**Algorithm 1:** Training Procedure of LaKD

---
**while** *Model not converges* **do**

    Sample trajectory $(\mathbf{X}^{obs}, \mathbf{X}^{gt})$ from dataset;

    **for** $m = 1$ *to* $M$ **do**

        Random mask to get $\mathbf{X}_m^{obs}$;

        Obtain predicted trajectories $\widehat{\mathbf{X}}_m$ by Equation (1);

        Compare the prediction performance of the current observation trajectory with the previous 'good' observation trajectory by Equation (2), and then determine the direction of knowledge distillation;

        Carry out knowledge distillation according to Equation (3);

        Calculate the total loss function $\mathcal{L}$ by Equation (9);

        Calculate importance scores of units $I_l^m$ by Equations (4) and (5);

        Calculate cumulative importance $I_l^{(\leq m-1)}$ by Equation (6);

        Constrain the gradient of units according to Equations (7) and (8);

        Update parameters using the AdamW optimizer.

    **end**

**end**

---

## A.2 Additional Experimental Results

In this section, we demonstrate our model's ability to process observed trajectories of arbitrary lengths using three metrics: minADE, minFDE, and MR. From the figures below, our method significantly outperforms other baselines in handling trajectory points of any length.

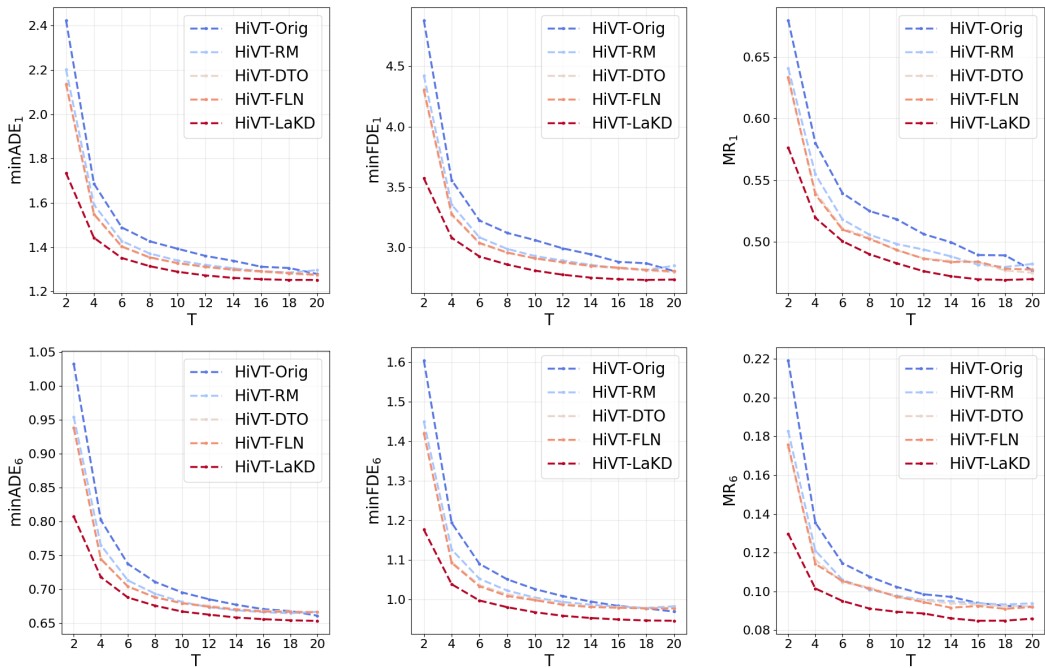

Figure 4: Comparison of Results Using HiVT as the backbone on the Argoverse 1 dataset.

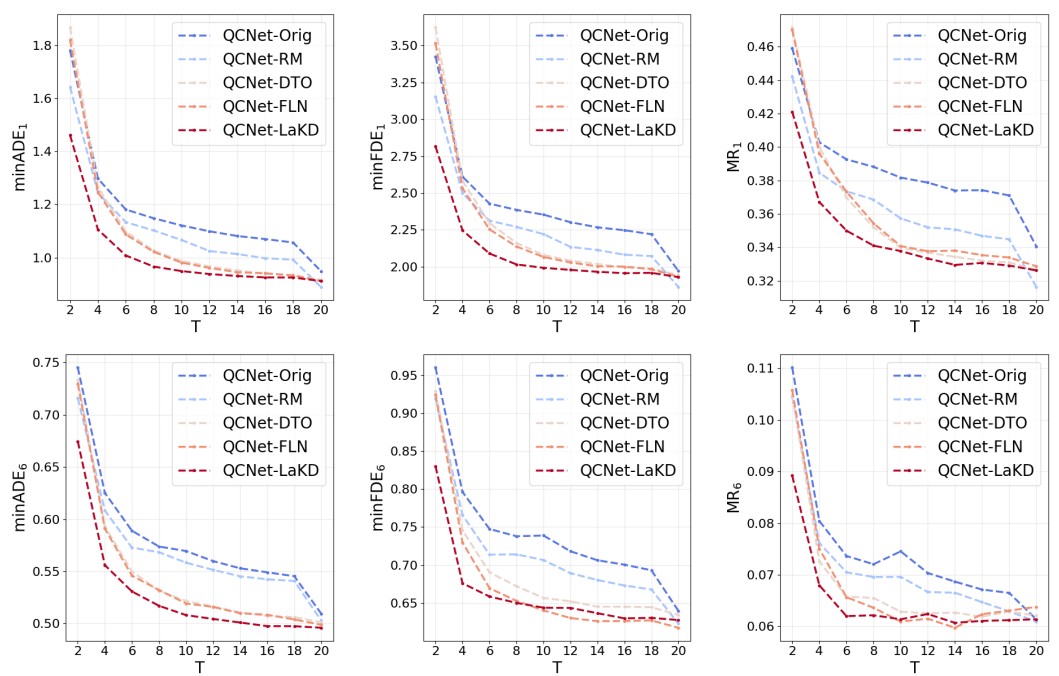

Figure 5: Comparison of Results Using QCNet as the backbone on the Argoverse 1 dataset.

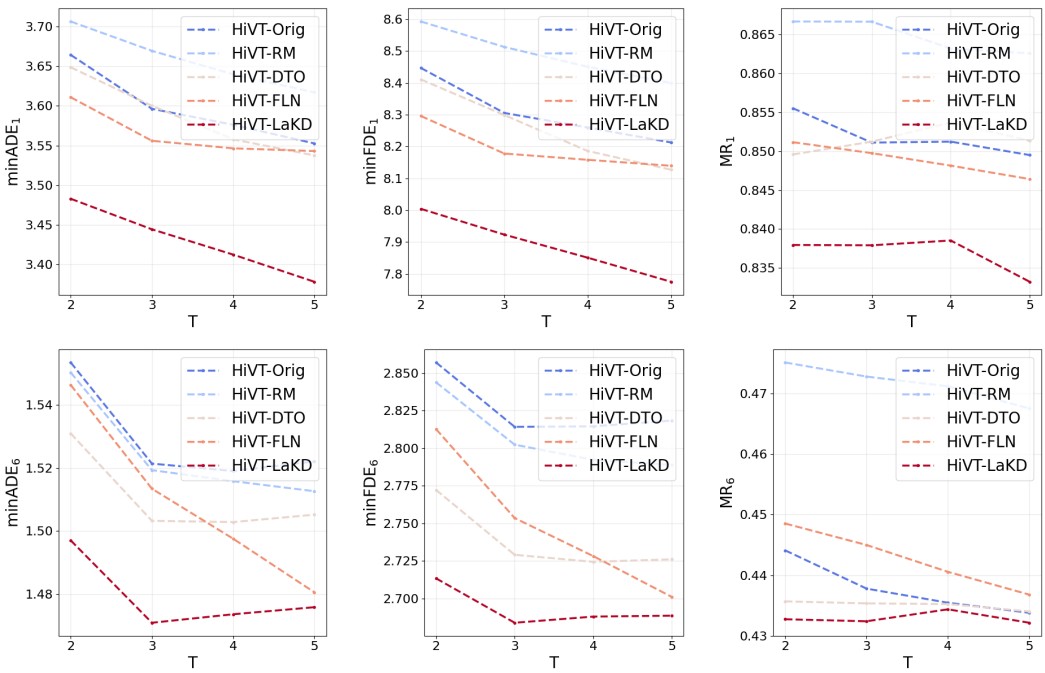

Figure 6: Comparison of Results Using HiVT as the backbone on the nuScenes dataset.

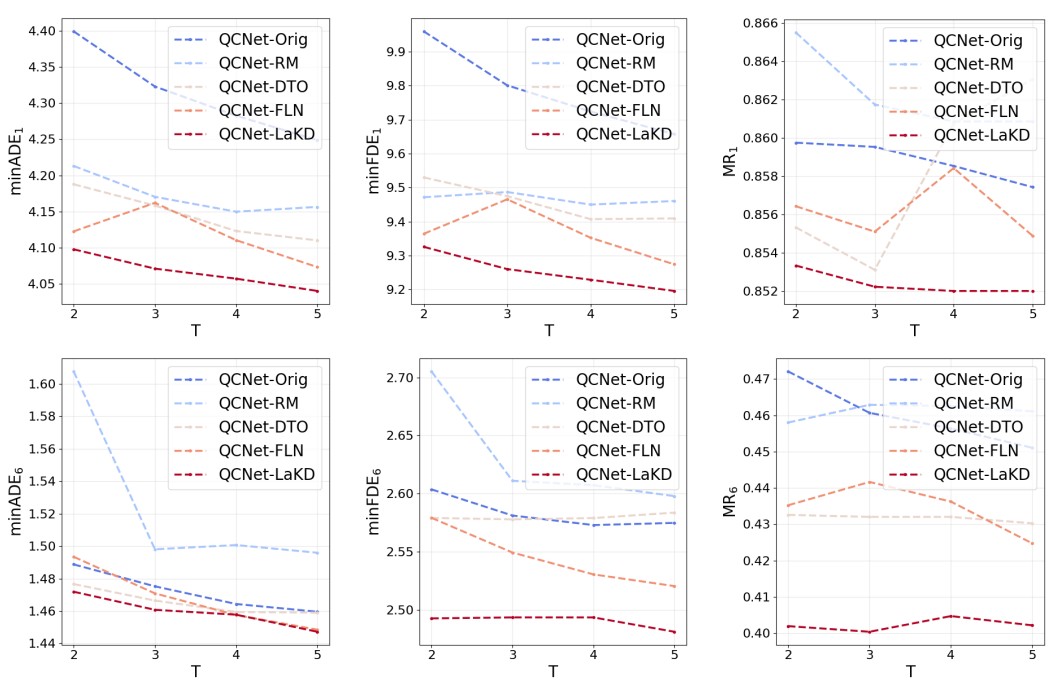

Figure 7: Comparison of Results Using QCNet as the backbone on the nuScenes dataset.

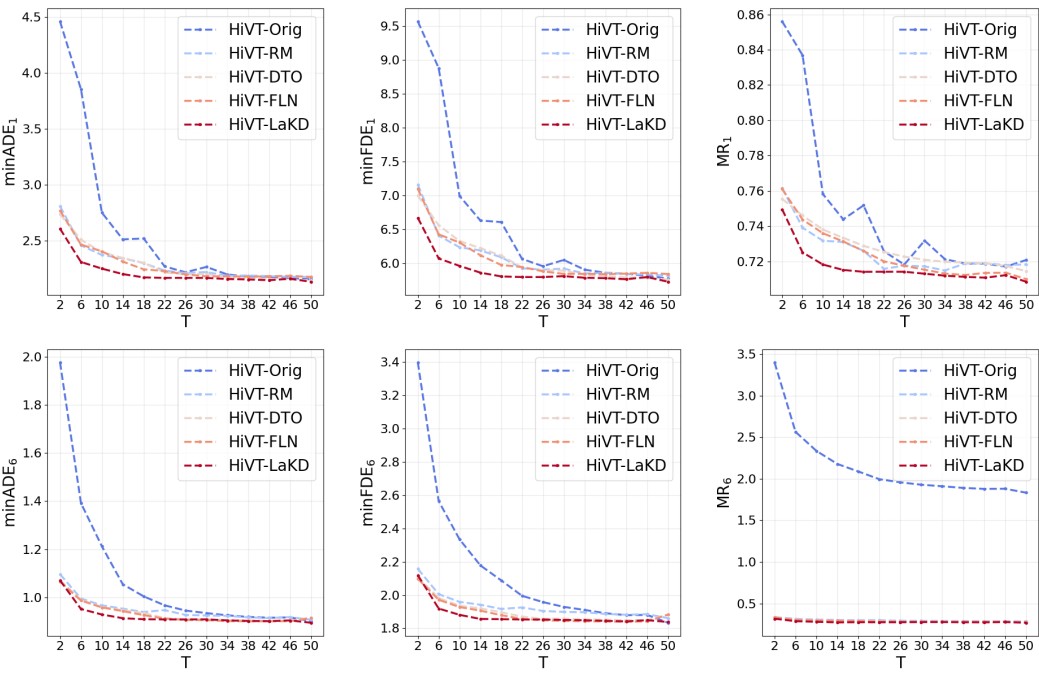

Figure 8: Comparison of Results Using HiVT as the backbone on the Argoverse 2 dataset.

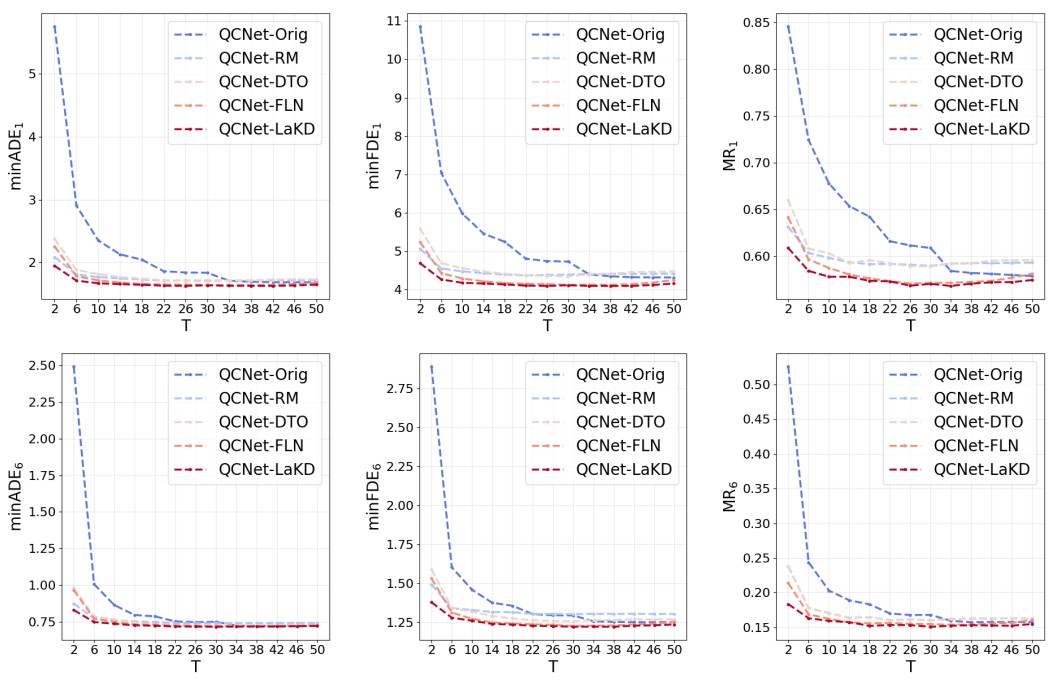

Figure 9: Comparison of Results Using QCNet as the backbone on the Argoverse 2 dataset.

## A.3 Limitations

In this work, we aim to distill knowledge from 'good' trajectory to 'bad' trajectory for improving the prediction performance from observations of any lengths. However, how to determine a 'good' or 'bad' trajectory is an open problem. Currently, we adopt a heuristic strategy by utilizing the distance between the predicted trajectory and the ground-truth trajectory. More complex strategies, such as reinforcement learning, are worth further exploration and investigation.

