# OpenReview forum: "LaKD: Length-agnostic Knowledge Distillation for Trajectory Prediction with Any Length Observations"
_NeurIPS.cc/2024/Conference — NeurIPS 2024 poster_

### Official Review · Reviewer_G7Yk · 2024-06-29

**Soundness:** 3
**Presentation:** 4
**Contribution:** 3
**Rating:** 7
**Confidence:** 5

**Summary:**

This paper presents the LaKD method to improve trajectory predictions for variable input observation lengths.

LaKD incorporates two key ideas. The first idea is dynamic length-agnostic knowledge distillation. During training time, for each training sample, they augment it by randomly masking the input observation with different lengths. Then, they calculate the prediction errors when using each observation length. They select the prediction with the lowest error as the teacher prediction, and they have the other predictions distill information from the teacher prediction through a KL divergence loss on their feature embeddings. This way, the network is trained to predict good trajectories when given any observation length.

In order to prevent the knowledge distillation process from affecting the performance of the teacher prediction, they propose a second idea to apply gradient weigths based on the importance of the neurons on the prediction prediction. For neurons that have high importance to the teacher prediction, they multiply their corresponding gradients with a smaller weight.

This method is a plug-and-play that can be applied to almost any trajectory prediction model. The authors applied the method to the popular HiVT model and QCNet model. They performed evaluations on Argoverse 1, Argoverse 2, and nuScenes datasets. They compared against a few standard baselines such as Original and Random Masking, as well as a similar work FlexiLength Network (FLN). The result shows their LaKD method achieves better performance than the baselines. The authors also performed ablation studies to demonstrate the contributions from components.

**Strengths:**

* The evaluation result is very thorough. The authors applied their method to two popular trajectory prediction models and performed evaluation on three public datasets.

* The proposed method achieves better performance than the baselines, including a very recent work from CVPR 2024.

* The paper is well-written and easy to follow.

**Weaknesses:**

* The performance improvement compared to the naive Random Masking baseline is not very significant. I doubt whether it's worth the complexity to use this method in practice.

**Questions:**

N/A

**Limitations:**

Yes

---

> ### Author Rebuttal · Authors · 2024-08-06
>
> We sincerely appreciate your time and efforts on evaluating our work. Here are my responses to your comments:
>
> > **Comment 1**: The performance improvement compared to the naive Random Masking baseline is not very significant. I doubt whether it's worth the complexity to use this method in practice.
>
> Thanks for your comment. Firstly, during testing, our model disables bi-directional self-distillation, and remains fully consistent with the backbone structure without requiring any additional inference time or resources. Additionally, our method enables existing models to effectively handle observed trajectories of varying lengths. Secondly, as shown in Table 1, LaKD outperforms Random Masking with relative improvements ranging from 3% to 11%. Additionally, Figures 4, 5, 6, and 7 in the appendix demonstrate that LaKD yields better results than Random Masking across various lengths of observed trajectories. For example, on the Argoverse1 dataset using HiVT as the backbone, LaKD achieves relative improvements of 21.2%/19.2%/10.0% at K=1 and 15.3%/18.8%/28.9% at K=6 over Random Masking in terms of minADE/minFDE/MR with 2-frame observed trajectories.

---

> > ### Comment · Reviewer_G7Yk · 2024-08-08
> > **Thank you for your response**
> >
> > Thank you for your response. I will keep my rating.

---

> > > ### Author Response · Authors · 2024-08-14
> > > **Official Comment by Authors**
> > >
> > > Thank you for your decision to maintain the current rating. I appreciate your time and feedback.

---

### Official Review · Reviewer_4u3J · 2024-07-11

**Soundness:** 3
**Presentation:** 3
**Contribution:** 3
**Rating:** 6
**Confidence:** 4

**Summary:**

To tackle the length-agnostic trajectory prediction problem, the authors are motivated to utilize knowledge learned from both longer and shorter trajectories. They propose a plug-and-play self-distillation framework for trajectory prediction, which can be integrated with many different off-the-shelf trajectory prediction models. Specifically, for a given long trajectory, the authors first randomly mask the trajectory into several different lengths, maintaining the same prediction horizon but with different history lengths. They then determine the direction of knowledge distillation based on the performance of the model with longer or shorter history. The better-performing trajectory length is used to distill knowledge into the intermediate features of represented from other lengths. Additionally, the authors introduce a soft masking method at neuron level, updating the distillation of important neurons slowly and vice versa. They test their framework on Argoverse1, nuScenes, and Argoverse2 datasets, using off-the-shelf models HiVT and QCNet, and demonstrate performance gains.

**Strengths:**

S1. The paper is generally well-written and easy to follow.

S2. The proposed plug-and-play method can work well with many different off-the-shelf model architectures, making it flexible. And the proposed method shows performance gains with different several recent trajectory prediction models.

**Weaknesses:**

W1. One constraint for most knowledge distillation scenarios is that for the input to the model being distilled and the model distilling, at least the information should be equivalent or nearly equivalent between the inputs to both models. I'm a bit confused about why the overall framework of bidirectional self-distillation works from an information theory / flow perspective. For my detailed confusion, please refer to my question section.

W2. For the qualitative analysis in Fig. 3, I could hardly understand why this shows that the proposed LRKD method is better than the other methods. In the above scenario, there doesn't seem to be a significant difference between the three predictions. In the below scenario, the proposed method is closer to the ground truth, but I do not fully understand why the ground truth behaves in that manner and how this is inferred by the model.

**Questions:**

Q1. Could the authors explain further what do they mean by “Knowledge Collision”?

Q2. May I ask how could a trivial degenerate solution, or a mode collapse, be avoided when conducting bi-directional self-distillation? I am aware that the authors try to classify the neurons in Section 3.4, but it seems to me that by the metrics as in Equation 4, the important neurons will get even more important length-agnostically, and therefore get distilled more both for long and for short trajectories.

Q3. A question about the overall framework is that when we distill the representation of a long history into a short history, the short history lacks information about the longer past. For example, when trajectory A and trajectory B have the same last several time steps but differ in that one turns left earlier while the other turns right, the information intrinsic to these trajectories is different. Let’s refer to the last time stamps of both A and B as a shorter C, as A and B share the same last steps. The longer horizon trajectory encoder of A can capture this information, but when it is distilled to the encoder with a shorter history C, this distinguished information of A is not seen. But the proposed framework enforces distilling this uncertain knowledge of A to the short history C, even if in a future potential test situation, the longer trajectory turns in the opposite direction in the past, like B, than seen in the training set. May I ask how is the proposed framework valid, considering a similar situation?

Q4. For Ablation Table 3, may I ask why the performance degrades when M is larger than 3? Intuitively, the performance should improve with a larger mask number?

**Limitations:**

Briefly discussed in Conclusion section.

---

> ### Author Rebuttal · Authors · 2024-08-06
>
> Thank you for your insightful review. The following are our responses to the points you have raised.
>
> > **Comment 1**: I'm a bit confused about why the overall framework of bidirectional self-distillation works from an information theory / flow perspective.
>
> Sorry for any confusion caused by our unclear presentation. Traditional knowledge distillation involves transferring knowledge from a deeper teacher model to a shallower student model given the same input, which is a form of model capability transfer. However, the bidirectional self-distillation in our LaKD first randomly masks the same trajectory to obtain trajectories of different lengths. Then, knowledge is transferred from the features of good trajectories to those of bad trajectories extracted by the same model, which is a form of data-level information transfer. We expect that this approach will enable trajectories of varying lengths to exhibit strong features for future trajectory prediction.
>
>
>
> > **Question 1**: Could the authors explain further what do they mean by “Knowledge Collision”?
>
> As mentioned above, since our framework uses a single shared encoder to extract feature representations for trajectories of different lengths and then performs data-level information transfer through this encoder, the feature representation of good trajectories can be disrupted when knowledge is transferred to bad trajectories through updates to the shared encoder. We refer to this problem as "Knowledge Collision". We will add more details in the final version.
>
>
>
> > **Comment 2**: For the qualitative analysis in Fig. 3, I could hardly understand why this shows that the proposed LRKD method is better than the other methods.
>
> Thanks for your comment.  Although it seems that there is no significant difference among the three predictions in the above scenario of Fig. 3, their results are 3.5891/6.0794/1, 1.4298/3.0295/1 and 0.6915/1.0974/0 in terms of minADE/minFDE/MR, respectively, indicating that our LaKD provides more accurate predictions. To more intuitively demonstrate the effectiveness of our LaKD method, we have included additional cases in the Figure 1 of the attached pdf to show that our method succeeds where others fail, as shown in the "global" response.
>
> Regarding the behavior of the ground truth in the below scenario of Fig. 3, we infer that the vehicle stops at the road ramp. We think this can be inferred by the features of the vehicle, e.g., speed, acceleration, etc.
>
>
>
>
>
>
>
> > **Question 2**: May I ask how could a trivial degenerate solution, or a mode collapse, be avoided when conducting bi-directional self-distillation?
>
> We apologize for the confusion caused by our unclear presentation. In our method, we design a dynamic soft-masking strategy to effectively conduct bi-directional self-distillation. During the training process, we set different soft-masking weights based on the importance of the units. This protects the more important units from significant damage that could affect the model's performance on trajectories of different lengths. However, these important units are still slowly updating, and their importance does not necessarily increase, as shown in Equation 7. Additionally, our dynamic soft-masking mechanism recalculates the importance of units whenever a new batch of data is encountered, avoiding excessive protection of any single unit. Furthermore, when the model's performance is poor, highly important units also need to be trained. Therefore, we lower their soft-masking weights at the early stages of training to prevent mode collapse, as shown in Equation 8. We will add more details in the final version.
>
>
>
> > **Question 3**: A question about the overall framework is that when we distill the representation of a long history into a short history, the short history lacks information about the longer past.
>
> Thank you for your insightful question. During the training process, we can perform our LaKD to distill knowledge from the longer trajectory A to the shorter trajectory C (Note that A and C belong to the same trajectory), allowing C to capture the intrinsic information of A and achieve similar prediction performance. By this means, we expect the shared encoder to extract feature representations for trajectories of different lengths as precisely as possible. During testing, we disable bi-directional self-distillation and only use the backbone module. When a new trajectory B (i.e., B and A belong to different trajectories) comes, we directly extract its features using the shared encoder for future trajectory prediction, which can capture the distinguishing information of B.
>
>
>
>
>
>
>
> > **Question 4**: For Ablation Table 3, may I ask why the performance degrades when M is larger than 3? Intuitively, the performance should improve with a larger mask number?
>
> Thanks for your comment. Since our method involves randomly masking historical trajectories M times in each training iteration and continues for a sufficient number of epochs, observation trajectories of all different lengths are seen during training, regardless of the value of M. Consequently, the model's performance does not significantly fluctuant as M changes, indicating that the model is not sensitive to M. This makes M easy to set in real-world scenarios.
>
> Regarding the performance degradation when M is larger than 3, this occurs because, despite our use of a dynamic soft-masking mechanism to prevent knowledge collision, the feature representation of good trajectories may be compromised when knowledge is transferred to bad trajectories through updates to the shared encoder. This compromise becomes more severe as M increases, resulting in reduced performance. We will add these in the final version.

---

> > ### Comment · Reviewer_4u3J · 2024-08-08
> > **Raised my rating from 5 to 6**
> >
> > Thank you so much for answering my questions! Most of my questions are well answered. It would be great if you could provide a more detailed explanation on question 3.

---

> > > ### Author Response · Authors · 2024-08-08
> > > **A more detailed explanation on question 3.**
> > >
> > > Thank you very much for recognizing our work. I will provide a more detailed explanation of Question 3, as follows:
> > > During the training process, we can perform our LaKD to distill knowledge from the longer trajectory A to the shorter trajectory C (Note that A and C belong to the same trajectory), allowing C to capture the intrinsic information of A and achieve similar prediction performance. By this means, we expect the shared encoder to extract feature representations for trajectories of different lengths as precisely as possible. Our Length-agnostic Knowledge Distillation framework will only enhance the model's ability to extract temporal features of trajectories, improving the model's capacity to capture every temporal change in the observed trajectory, without reducing the model's generalization ability.
> > >
> > > During testing, we disable bi-directional self-distillation and only use the backbone module. When a new unseen trajectory B (i.e., B and A belong to different trajectories) comes, the model does not predict based on the previously seen trajectories A and C, but instead directly extracts its features using the shared encoder for future trajectory prediction. The model can capture the distinguishing information of B, recognize B's turning behavior, and make predictions based solely on B's observed trajectory, without being influenced by A and C.

---

### Official Review · Reviewer_Rudm · 2024-07-16

**Soundness:** 3
**Presentation:** 3
**Contribution:** 3
**Rating:** 7
**Confidence:** 3

**Summary:**

The paper presents a length-agnostic knowledge distillation framework, which is motivated from knowledge transfer among trajectories of different lengths. The authors address knowledge conflicts during distillation from a dynamic soft-masking mechanism. The evaluation is conducted using Argoverse 1, nuScenes, and Argoverse 2 datasets.

**Strengths:**

- The problem is clearly stated and the motivation is logical.
- Dynamic knowledge transfer between trajectories of varying lengths sounds interesting.
- It seems that the soft-masking strategy reasonably addresses the issue of knowledge collision as proposed.

**Weaknesses:**

- Lacks a proper qualitative results.
- Limited choice of backbone models.
See below for details.

**Questions:**

- The qualitative results do not clearly demonstrate the scenarios where the authors have been motivated. As shown in Figure 3, there is no difference between (b) and (c), which makes me doubt about the contribution of this work. Not just closer to the ground-truth, it is strongly suggested where the proposed method works while others fail.
- Different backbone models other than HiVT and QCNet would be needed to claim its generality.

**Limitations:**

The paper does not provide any failure cases or insights into the limitations.

---

> ### Author Rebuttal · Authors · 2024-08-06
>
> We truly appreciate the reviewer of the constructive feedback. In light of these insightful comments, we would like to address them by providing the following clarifications.
>
> > **Question 1**: The qualitative results do not clearly demonstrate the scenarios where the authors have been motivated. As shown in Figure 3, there is no difference between (b) and (c), which makes me doubt about the contribution of this work. Not just closer to the ground-truth, it is strongly suggested where the proposed method works while others fail.
>
> Thanks for raising this concern which helps us to clarify the contribution of our work. Due to the limited time during the submission phase, we simply visualized the results to demonstrate that our method can more accurately predict future trajectories. Although it seems that there is no significant difference among the three predictions in the above scenario of Fig. 3, their results are 3.5891/6.0794/1, 1.4298/3.0295/1 and 0.6915/1.0974/0 in terms of minADE/minFDE/MR, respectively, indicating that our LaKD provides more accurate predictions. Following your suggestion, we have added more results to demonstrate that our method works while others fail, as shown in the Figure 1 of the attached pdf in the "global" response. Thanks again for the insightful comments, and we will add these results in the final version.
>
>
>
>
>
> > **Question 2**: Different backbone models other than HiVT and QCNet would be needed to claim its generality.
>
> Thanks for the comment. We chose to use HiVT and QCNet as our backbones in our paper because they are the most advanced models in the field of trajectory prediction and have recently ranked highly on the Argoverse dataset leaderboard. Based on your helpful suggestion, we conducted experiments using two other typical trajectory prediction models, TNT [1] and Vectornet [2], as our backbones to show the generality of our method. The results are as follows:
>
> |||||||||
> |-|-|-|-|-|-|-|-|
> |Dataset|Methods||K=1|||K=6||
> |||$\mathrm{min\overline{ADE}}$|$\mathrm{min\overline{FDE}}$|$\mathrm{\overline{MR}}$|$\mathrm{min\overline{ADE}}$|$\mathrm{min\overline{FDE}}$|$\mathrm{\overline{MR}}$|
> |Argoverse 1|VectorNet-Orig|3.5335|7.8259|0.8267|2.1173|3.6945|0.6751|
> ||VectorNet-RM|1.7815|3.8589|0.6412|1.1016|1.9880|0.3531|
> ||VectorNet-DTO|1.7523|3.8065|0.6409|1.0126|1.8096|0.3189|
> ||VectorNet-FLN|1.7334|3.7027|0.6244|1.0088|1.7994|0.3132|
> ||VectorNet-LaKD|1.6003|3.4628|0.5917|0.9933|1.7546|0.3016|
> |Argoverse 1|TNT-Orig|3.7318|7.7174|0.8600|1.8255|2.9818|0.4081|
> ||TNT-RM|2.8629|6.3561|0.7925|1.1946|2.1685|0.3116|
> ||TNT-DTO|2.7280|6.1935|0.7858|1.1242|2.0630|0.3084|
> ||TNT-FLN|2.4241|5.5038|0.7593|1.0692|1.9485|0.2714|
> ||TNT-LaKD|2.2174|5.0279|0.6974|1.0172|1.8571|0.2456|
>
>
>
>
> The experimental results show that our method still achieves the best performance when using TNT and Vectornet as our backbones. We will add these results in the final version.
>
>
>
>
>
> > **Limitations**: The paper does not provide any failure cases or insights into the limitations.
>
> Thanks for your comment. Due to the limitations of the paper's length, we have analyzed our algorithm's limitations in the appendix as :
>
> In this work, we aim to distill knowledge from 'good' trajectory to 'bad' trajectory for improving the prediction performance from observations of any lengths. However, how to determine a 'good' or 'bad' trajectory is an open problem. Currently, we adopt a heuristic strategy by utilizing the distance between the predicted trajectory and the ground-truth trajectory. More complex strategies, such as reinforcement learning, are worth further exploration and investigation.
>
>
>
> [1] Zhao H, et al. "Tnt: Target-driven trajectory prediction" PMLR 2021.
>
> [2] Gao J, et al. "Vectornet: Encoding hd maps and agent dynamics from vectorized representation" CVPR 2020.

---

> > ### Comment · Reviewer_Rudm · 2024-08-13
> >
> > Thanks for providing the response. They are clear now. Please add new figure and results to the final version.

---

> > > ### Author Response · Authors · 2024-08-14
> > > **Official Comment by Authors**
> > >
> > > Thank you very much for the score improvement and your constructive feedback. We will further polish the paper in the final revision. Thank you!

---

### Author Rebuttal · Authors · 2024-08-06

We thank all the reviewers for their insightful and constructive feedback. We really appreciate the reviewers thought our work to be "well-motivated" (Rudm, 4u3J, G7Yk), "easy to follow" (Rudm, 4u3J, G7Yk), "well-written" (Rudm, 4u3J, G7Yk), "novel" (Rudm, 4u3J, G7Yk), "a promising/good topic" (Rudm, 4u3J, G7Yk), "effective in experiments" (Rudm, 4u3J, G7Yk), "sufficient experiments" (4u3J, G7Yk), "tackles an important problem" (Rudm, 4u3J, G7Yk), "insightful and sound" (Rudm, 4u3J, G7Yk), "with solid theoretical part" (Rudm, G7Yk).

We have made point-to-point response to the comments of each reviewer. Additionally, we provide further qualitative results in the attached PDF to address the reviewers' concerns. Once again, we thank all reviewers for their insightful comments which are very helpful for improving the quality of our paper.

---

### Decision · Program_Chairs · 2024-09-25

**Decision:**

Accept (poster)

**Comment:**

All reviewers have positively rated the paper. The AC agrees with the reviewers. Here are some additional comments to potentially improve the final version:
1-  Please discuss if LaKD may face scalability issues with large datasets due to the intensive computation required for importance scores and gradient adjustments.
2- Dynamic soft-masking may not fully resolve knowledge conflicts, as its effectiveness depends on accurately determining neuron importance, which can be challenging with diverse data patterns.

Thanks.